# Neuron-Level Sequential Editing for Large Language Models

## Abstract

This work explores sequential model editing in large language models (LLMs), a critical task that involves modifying internal knowledge within LLMs continuously through multi-round editing, each incorporating updates or corrections to adjust the model's outputs without the need for costly retraining. Existing model editing methods, especially those that alter model parameters, typically focus on single-round editing and often face significant challenges in sequential model editing-most notably issues of model forgetting and failure. To address these challenges, we introduce a new model editing method, namely **N**euron-level **S**equential **E**diting (NSE), tailored for supporting sequential model editing. Specifically, we optimize the target layer's hidden states using the model's original weights to prevent model failure. Furthermore, we iteratively select neurons in multiple layers for editing based on their activation values to mitigate model forgetting. Our empirical experiments demonstrate that NSE significantly outperforms current modifying parameters model editing methods, marking a substantial advancement in the field of sequential model editing. Our code is released on https://anonymous.4open.science/r/NSE-0A8D/.

## 1 Introduction

Large language models (LLMs) have demonstrated remarkable capabilities in storing extensive factual knowledge during pre-training and recalling this information during inference (Brown et al., 2020; Petroni et al., 2019; Roberts et al., 2020). However, as real-world knowledge continuously evolves, the information within these models can become outdated or incorrect (Cao et al., 2021; Mitchell et al., 2022a). Retraining LLMs to incorporate new information is often prohibitively costly (Mitchell et al., 2022b; Meng et al., 2022). Consequently, recent years have witnessed a surge in *model editing* methods focusing on modifying specific knowledge without the complete retraining process. Specifically, they first identify the crucial layers for the target knowledge by calculating their casual effect on output. Then, by updating the weights of these layers, they manipulate these layers' hidden states to modify the final output, enabling LLMs to adapt seamlessly to dynamic real-world information (Meng et al., 2023; Hartvigsen et al., 2023).

While current direct model editing methods prove effective for single-round modifications, real-world applications demand a continual learning process where models must retain previous edits during subsequent modifications (Yao et al., 2023). This has led to the concept of *sequential model editing*, which necessitates performing multiple, consecutive edits on models. However, current direct model editing methods pose significant risks in this context (Meng et al., 2022; 2023). The primary risk is *model forgetting*, where cumulative changes in parameters from consecutive edits cause the model to forget previously modified knowledge, thereby degrading overall performance (Gupta et al., 2024a). For instance, as illustrated in Figure 1 (a), after editing the model with new knowledge about "The cat", the LLM forgets previously edited knowledge about "The latest Olympic".

Furthermore, the second risk is *model failure*, where excessive edits impair the model's ability to generate coherent text. Worse still, this impairment may lead to model collapse potentially, characterized by producing irrelevant, repetitive, or nonsensical text, as illustrated in Figure 1 (a). In sight of this, recent researches, such as memory-based methods (Mitchell et al., 2022b; Hartvigsen et al., 2023; Das et al., 2024), have attempted to address these challenges by preserving LLM

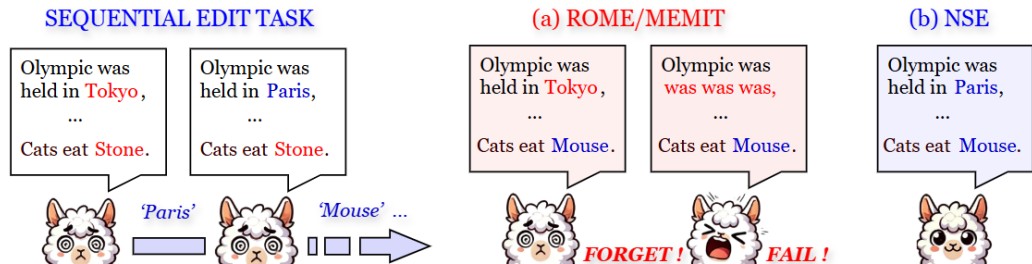

Figure 1: Example of sequential editing. (a) shows model forgetting and model failure issues in sequential editing using ROME/MEMIT, while (b) shows the accurate editing capabilities of our method without such issues.

parameters after each edit. However, the increasing storage requirements as the number of edits grows significantly limit the practicality of these methods.

To tackle these challenges, we introduce a new model editing method, termed **N**euron-level **S**equential **E**diting (**NSE**). Specifically, to address the model failure, NSE uses weights rewinding for value computation by preserving the model's original weights as a significant reference when manipulating the hidden states of the crucial layers. This process can effectively mitigate the impairment of previous knowledge accumulated over various edits. Furthermore, to address model forgetting, NSE selectively collects "influential neurons" for weights updating by sorting the neuron activation within the crucial layers, rather than updating all weights in critical layers as in previous work (Meng et al., 2022; 2023). This selective modification maximizes the protection of model functionality from being degraded. Additionally, for large-scale LLMs containing numerous neurons, an iterative multi-layer editing is introduced to streamline the neuron selection process, enabling NSE to achieve massive knowledge updates effectively in a single editing.

Through theoretical analysis and extensive experiments conducted on the GPT2-XL (1.5B) (Radford et al., 2019), GPT-J (6B) (Wang & Komatsuzaki, 2021) and Llama3 (8B), we validate the effectiveness and efficiency of our NSE. Compared to the current model editing methods (*e.g.*, Fine-tuning (Zhu et al., 2020), MEND (Mitchell et al., 2022a), ROME (Meng et al., 2022), MEMIT (Meng et al., 2023) and GRACE (Hartvigsen et al., 2023)), NSE shows substantial improvements *w.r.t* five commonly used metrics such as specificity and consistency.

## 2 PRELIMINARY

### 2.1 AUTOREGRESSIVE LANGUAGE MODEL

An autoregressive language model predicts the next token in a sequence based on the tokens that have come before it. Given a $L$-layer transformer model and an input sequence $x = (x_0, x_1, \ldots, x_T)$, the model aims to predict the next token in the sequence. The probability of the next token $x_{t+1}$ is given by:

$$\mathbb{P}(x_{t+1} \mid x_0, x_1, \ldots, x_t) = \text{Softmax}(\mathbf{W}_e \mathbf{h}_t^N), \tag{1}$$

where $\mathbf{W}_e$ represents embedding matrix, and $\mathbf{h}_t^N$ represents the final hidden state at the topmost layer $N$. The hidden state $\mathbf{h}_t^l$ at layer $l$ is calculated as:

$$\begin{aligned} \mathbf{h}_t^l(x) &= \mathbf{h}_t^{l-1}(x) + \mathbf{a}_t^l(x) + \mathbf{v}_t^l(x), \\ \mathbf{a}_t^l &= \text{attn}^l(\mathbf{h}_0^{l-1}, \mathbf{h}_1^{l-1}, \ldots, \mathbf{h}_t^{l-1}), \\ \mathbf{v}_t^l &= \mathbf{W}_{\text{out}}^l \sigma(\mathbf{W}_{\text{in}}^l \gamma(\mathbf{h}_t^{l-1} + \mathbf{a}_t^l)), \end{aligned} \tag{2}$$

where $\mathbf{a}_t^l$ represents the output of the attention block and $\mathbf{v}_t^l$ represents the output of the FFN layer. $\mathbf{W}_{\text{in}}^l$ and $\mathbf{W}_{\text{out}}^l$ are weight matrices, $\sigma$ is non-linear activation function, and $\gamma$ represents layernorm.

### 2.2 SEQUENTIAL MODEL EDITING

Sequential model editing aims to refine a pre-trained model $f_{\boldsymbol{\theta}_0}$ continuously through multiple edits, each incorporating updates or corrections to adjust the model's outputs (Huang et al., 2023;

Hartvigsen et al., 2023; Li et al., 2024). Formally, each batch of sequential edits involves a series of facts $(s, r, o)$ in the form of (subject $s$, relation $r$, object $o$) (*e.g.*, $s$="The latest Olympic", $r$="was held in", $o$="Paris") for each edit iteration $t$, specifying the desired responses for that round. After the $t$-th edit, the updated model $f_{\boldsymbol{\theta}_t}$, built on its predecessor $f_{\boldsymbol{\theta}_{t-1}}$, is optimized to accurately produce the target outputs for the relevant inputs $\mathbb{D}_{edit_t}$, while maintaining accuracy on inputs outside the current edit scope. This ensures that the model not only adapts to new requirements but also preserves effectiveness across the whole process.

More formally, the editing function $h$ for each edit $t$ is defined as $f_{\boldsymbol{\theta}_t} = h(f_{\boldsymbol{\theta}_{t-1}}, \mathbb{D}_{edit_t})$, applying necessary updates based on the specific edits. Following prior studies (Meng et al., 2022; 2023), we encourage the editing function to satisfy the following goals:

- **Efficacy.** For all inputs in any previous editing rounds up to the current $t$-th round, the updated model $f_{\boldsymbol{\theta}_t}$ consistently maintains the target outputs:

$$f_{\boldsymbol{\theta}_t}((s, r)) = o, \quad \forall (s, r, o) \in \bigcup_{j=1}^{t} \mathbb{D}_{edit_j}. \tag{3}$$

- **Generalization.** For any inputs equivalent to the edited input $(s, r)$, denoted by $N((s, r))$, the updated model $f_{\boldsymbol{\theta}_t}$ consistently outputs the intended result $o$ for all edits up to the current round:

$$f_{\boldsymbol{\theta}_t}(N((s, r))) = o, \quad \forall (s, r, o) \in \bigcup_{j=1}^{t} \mathbb{D}_{edit_j}. \tag{4}$$

- **Specificity.** The updated model $f_{\boldsymbol{\theta}_t}$ retains the outputs from its initial model $f_{\boldsymbol{\theta}_0}$ for all inputs that have not been edited in any round up to the current one:

$$f_{\boldsymbol{\theta}_t}((s, r)) = f_{\boldsymbol{\theta}_0}((s, r)), \quad \forall (s, r, o) \notin \bigcup_{j=1}^{t} \mathbb{D}_{edit_j}. \tag{5}$$

## 2.3 Previous Model Editing Method

Given a knowledge fact tuple $(s_i, r_i, o_i)$, the objective is to edit the output of the LLM such that prompting it with $(s_i, r_i)$ results in $o_i$. Following ROME (Meng et al., 2022) and MEMIT (Meng et al., 2023), we treat the weights of the Transformer's (Vaswani et al., 2017) FFN layer as a linear associative memory. That is, the linear operations within the FFN layer can be viewed as key-value storage for information retrieval (Kohonen, 1972; Anderson, 1972).

Given the weights $\boldsymbol{W}_{\text{in}}^l$ of the $l$-th FFN layer when prompted with $(s_i, r_i)$, we identify the activation output of the last subject token $S$ as the key $\boldsymbol{k}_i^l$. Hereafter, this key $\boldsymbol{k}_i^l$ is processed through the output weights $\boldsymbol{W}_{\text{out}}^l$, producing the value $\boldsymbol{v}_i^l$. In the context of sequential model editing, we start with an initial set of key-value associations for knowledge facts stored in the $l$-th FFN layer, denoted respectively by $K_0 = \{\boldsymbol{k}_i\}_{i=1}^n$ and $V_0 = \{\boldsymbol{v}_i\}_{i=1}^n$. Hereafter, we aim to introduce $m$ new key-value associations, denoted as $\boldsymbol{K}_1 = \{\boldsymbol{k}_i\}_{i=n+1}^{n+m}$ and $\boldsymbol{V}_1 = \{\boldsymbol{v}_i\}_{i=n+1}^{n+m}$, while retaining all existing associations unchanged. Drawing on prior work (Meng et al., 2023), the optimization objective is as follows:

$$\boldsymbol{\Delta}^* = \arg\min_{\boldsymbol{\Delta}}(\|(\boldsymbol{W} + \boldsymbol{\Delta})\boldsymbol{K}_1 - \boldsymbol{V}_1\|^2 + \|(\boldsymbol{W} + \boldsymbol{\Delta})\boldsymbol{K}_0 - \boldsymbol{V}_0\|^2), \tag{6}$$

where $\boldsymbol{W}$ represents the weights of $\boldsymbol{W}_{\text{out}}$ in the target FFN layer, $\Delta$ denotes the update to $W$, and $V_1$ can be directly trained using the fine-tuning loss predicted by the model through backpropagation. Since $\boldsymbol{K}_0$ and $\boldsymbol{V}_0$ represent the retained knowledge in LLMs, we can express this relationship as $\boldsymbol{W}\boldsymbol{K}_0 = \boldsymbol{V}_0$. Therefore, we can obtain the closed-form solution of Eqn. 6 using the least squares method (Lang, 2012):

$$\boldsymbol{\Delta}^* = \boldsymbol{R}\boldsymbol{K}_1^T \left(\boldsymbol{K}_0\boldsymbol{K}_0^T + \boldsymbol{K}_1\boldsymbol{K}_1^T\right)^{-1}, \tag{7}$$

where $\boldsymbol{R} = \boldsymbol{V}_1 - \boldsymbol{W}\boldsymbol{K}_1$. Additionally, MEMIT focuses on editing a specific set of layers denoted as $\mathcal{R} = \{l_0 - |\mathcal{R}| + 1, \ldots, l_0\}$. The required the weight update $\boldsymbol{\Delta}^l$ for layer $l \in \mathcal{R}$ is expressed as:

$$\boldsymbol{\Delta}^l = \boldsymbol{R}^l\boldsymbol{K}_1^{l\,T} \left(\boldsymbol{K}_0^l\boldsymbol{K}_0^{l\,T} + \boldsymbol{K}_1^l\boldsymbol{K}_1^{l\,T}\right)^{-1}, \tag{8}$$

where $\boldsymbol{R}^l = \frac{\boldsymbol{R}^{l_0}}{l_0 - l + 1}$. These modifications are implemented sequentially, starting from the lower layers and progressing to the upper layers.

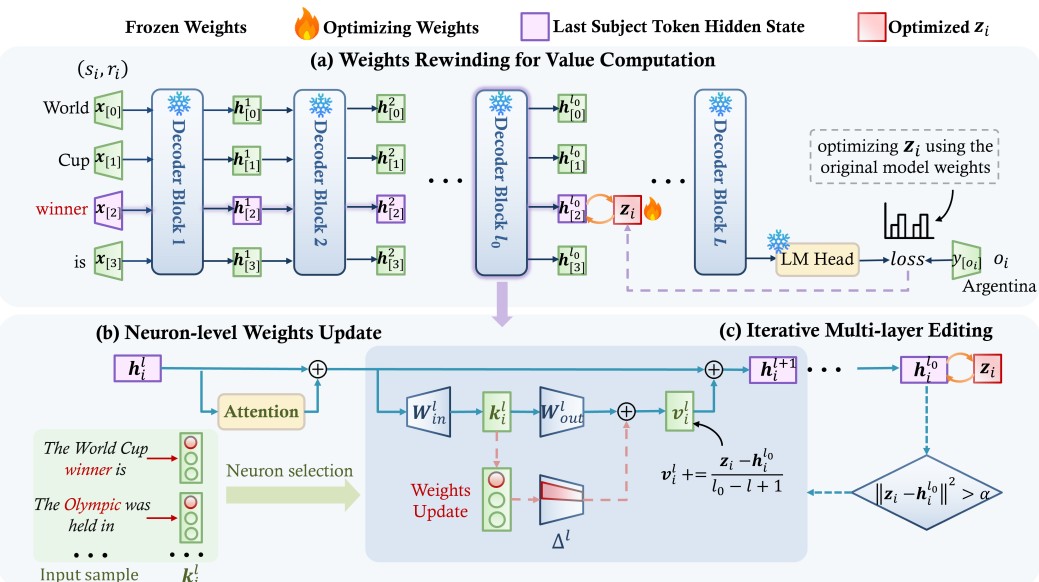

Figure 2: Overview of sequential model editing with NSE. (a) describes the process of weights rewinding for value computation. (b) illustrates the neuron selection and neuron-level weights update. (c) shows the process of iterative multi-layer editing.

## 3 METHODOLOGY

In this section, we introduce NSE, a method tailored for sequential model editing, as illustrated in Figure 2. Initially, we identify the values computing method in Section 3.1. Subsequently, in Section 3.2, we detail the editing method that selectively filters certain neurons for corresponding parameter updates. Finally, in Section 3.3, we introduce the approach of iterative multi-layer editing.

### 3.1 WEIGHTS REWINDING FOR VALUE COMPUTATION

First of all, our primary goal is to find a hidden vector that encodes the new association $(s_i, r_i, o_i)$ and replaces the value $v_i^l$ in $l$-th layer as described in Section 2.3. In practical implementation, as shown in Figure 2 (a), we optimize $\delta_i$ through gradient descent by maximizing the probability of the model outputting $o_i$ to compute $z_i = h_i^l + \delta_i$, where $h_i^l$ denotes the hidden state of the LLM at layer $l$. And the value $v_i^l$ can be computed as $v_i^l += \delta_i$. In the process of sequential editing, we observed that using the updated model parameters $f_{\theta_t}$ to compute $z_i$ after each edit leads to significant model degradation over multiple edit rounds. This indicates that the cumulative parameter updates from each editing round can lead to a shift in value computation. Conversely, using the original model parameters $f_{\theta_0}$ to compute $z_i$ effectively prevents this issue. Hence, we propose a weights rewinding method for value computation, which is based on the preserve initial model weights $f_{\theta_0}$ to ensure that $z_i$ is computed using $f_{\theta_0}$ for each edit. The optimization objective is as follows:

$$z_i = h_i^l + \arg\min_{\delta_i}(-\log \mathbb{P}_{f_{\theta_0}(h_i^l += \delta_i)}[o_i \mid (s_i, r_i)]), \tag{9}$$

where $f_{\theta_0}(h_i^l += \delta_i)$ represents the original model with $h_i^l$ updated to $h_i^l + \delta_i$. Subsequently, the value $v_i^l$ can be updated as $v_i^l += z_i - h_i^l$, which will be used to support subsequent model edits. Note that when calculating $z_i$ using the original model parameters $f_{\theta_0}$, it is sufficient to save only the weight matrix $W_{\text{out}}$ that needs to be updated, rather than storing the entire model parameters and we use the original weights only during the computation of $z_i$.

### 3.2 NEURON-LEVEL WEIGHTS UPDATING

In Section 3.1, we primarily discussed the method for calculating $z_i$ to replace the value in the target layer. In this section, we will specifically elaborate on how to utilize the computed value for neuron-level weight updates, as shown in Figure 2 (b).

Based on previous works, it is established that neurons in FFN contain abundant information (Dai et al., 2022; Wang et al., 2022; Schwettmann et al., 2023; Pan et al., 2023). Hence, we selectively optimize a subset of neurons rather than alter the entire weight matrix for each edit. Specifically, for a given knowledge fact $(s_i, r_i, o_i)$, we utilize the activation values $\boldsymbol{k}_i$ of the neurons and compute scores $\mathbf{Q}_i = |\boldsymbol{k}_i|$. Neurons are ranked based on these scores, and a subset is chosen such that the cumulative score of selected neurons surpasses a predetermined percentage of the total score:

$$\mathcal{I} = \underset{\mathcal{I} \subseteq \{1, \ldots, N\}}{\arg\min} |\mathcal{I}| \quad \text{s.t.} \quad \sum_{j \in \mathcal{I}} \boldsymbol{Q}_{ij} \geq p \times \sum_{j=1}^{N} \boldsymbol{Q}_{ij}, \tag{10}$$

where $\boldsymbol{Q}_{ij}$ represents the score of the $j$-th neuron and $\mathcal{I}$ is the set of indices for selected neurons.

Given the need for batch editing, which entails editing multiple knowledge facts simultaneously, each possibly corresponding to different neuron sets, we compute the sum of neuron scores corresponding to all samples in a batch to derive a new neuron score, which is then used for selecting neurons.

Hereafter, we introduce how to update $\boldsymbol{W}$ (i.e., $\boldsymbol{W}_{\text{out}}$) by the selected neurons set $\mathcal{I}$. Let $\hat{\boldsymbol{W}}$ and $\hat{\boldsymbol{\Delta}}$ denote the submatrices of $\boldsymbol{W}$ and $\boldsymbol{\Delta}$, respectively, selected according to the indices in the set $\mathcal{I}$. Our objective is to modify only a subset of neuronal parameters by altering specific rows of weights in $\boldsymbol{W}$ to achieve the optimization outlined in Eqn. 6. And we can transform Eqn. 6 accordingly:

$$\hat{\boldsymbol{\Delta}}^* = \underset{\hat{\boldsymbol{\Delta}}}{\arg\min}(\left\|(\hat{\boldsymbol{W}} + \hat{\boldsymbol{\Delta}})\hat{\boldsymbol{K}}_1 - \boldsymbol{V}_1\right\|^2 + \left\|(\hat{\boldsymbol{W}} + \hat{\boldsymbol{\Delta}})\hat{\boldsymbol{K}}_0 - \boldsymbol{V}_0\right\|^2), \tag{11}$$

where $\hat{\boldsymbol{K}}_0$ and $\hat{\boldsymbol{K}}_1$ are the submatrices of $\boldsymbol{K}_0$ and $\boldsymbol{K}_1$ formed by selecting elements indexed by $\mathcal{I}$ respectively. Based on the method of minimal squared error, we solve Eqn. 11 as follows:

$$\hat{\boldsymbol{\Delta}}^* = \hat{\boldsymbol{R}}\hat{\boldsymbol{K}}_1^T \hat{\boldsymbol{C}}^{-1}, \tag{12}$$

where $\hat{\boldsymbol{R}} = \boldsymbol{V}_1 - \hat{\boldsymbol{W}}\hat{\boldsymbol{K}}_1$ and $\hat{\boldsymbol{C}} = \hat{\boldsymbol{K}}_0\hat{\boldsymbol{K}}_0^T + \hat{\boldsymbol{K}}_1\hat{\boldsymbol{K}}_1^T$, and following MEMIT (Meng et al., 2023), $\boldsymbol{K}_0\boldsymbol{K}_0^T$ is estimated by $\lambda\mathbb{E}\left[\boldsymbol{k}\boldsymbol{k}^T\right]$, where $\lambda$ is a hyperparameter that balances the weights between new knowledge and preserved knowledge. The submatrix $\hat{\boldsymbol{K}}_0\hat{\boldsymbol{K}}_0^T$ is then obtained by selecting the rows and columns indexed by $\mathcal{I}$ from $\boldsymbol{K}_0\boldsymbol{K}_0^T$. Additionally, during the continuous editing process, the newly edited knowledge from each round will become the old knowledge for the next round. Therefore, after each round, we will add the newly edited knowledge into $\boldsymbol{K}_0\boldsymbol{K}_0^T$.

### 3.3 ITERATIVE MULTI-LAYER EDITING

As described in Section 2.3, MEMIT propagates edits through the layers by computing the value $\boldsymbol{v}_i^l$ as $\boldsymbol{v}_i^l += \frac{\boldsymbol{\delta}_i}{l_0 - l + 1}$ $(l \in \mathcal{R})$ (Meng et al., 2023; Gupta et al., 2024b). Here, $\boldsymbol{\delta}_i = \boldsymbol{z}_i - \boldsymbol{h}_i^{l_0}$ represents the residual difference. The fundamental purpose is that as each layer is updated, the hidden state $\boldsymbol{h}_i^{l_0}$ progressively approaches the target $\boldsymbol{z}_i$, thereby diminishing the residual $\boldsymbol{\delta}_i$. Detailed analyses can be found in Appendix E. However, due to errors in the fitting process, some knowledge proves difficult to edit, resulting in $\boldsymbol{v}_i^{l_0}$ not sufficiently approximating $\boldsymbol{z}_i$, and consequently leading to editing failures.

Therefore, we propose iterative multi-layer editing to refine the multi-layer editing approach in MEMIT by iteratively selecting neurons to edit multiple layers, as depicted in Figure 2 (c). Specifically, considering that some knowledge is difficult to edit such that the corresponding value $\boldsymbol{v}_i^{l_0}$ cannot sufficiently approximate the optimized target value $\boldsymbol{z}_i$, we employ iterative multi-layer editing. After each round of multi-layer editing, we filter the knowledge samples in the current batch based on $\|\boldsymbol{z}_i - \boldsymbol{h}_i^{l_0}\|^2$. If $\|\boldsymbol{z}_i - \boldsymbol{h}_i^{l_0}\|^2 < \alpha$, the knowledge sample is considered successfully edited. Conversely, if $\|\boldsymbol{z}_i - \boldsymbol{h}_i^{l_0}\|^2 > \alpha$, the sample is deemed not yet successfully edited. The hyperparameter $\alpha$, which is set differently for various LLMs, determines the threshold for editing success. These unedited samples are then filtered out to form a new batch for further multi-layer editing, repeating this process until all knowledge samples in the batch meet $\|\boldsymbol{z}_i - \boldsymbol{h}_i^{l_0}\|^2 < \alpha$ or the iteration limit is reached. This iterative editing approach significantly enhances the success rate of sample edits, enabling NSE to effectively achieve massive knowledge updates in a single editing session. More details of the NSE algorithm are provided in Appendix A.

Table 1: Comparison of NSE with existing methods on the sequential model editing task. *Eff.*, *Gen.*, *Spe.*, *Flu.* and *Consis.* denote Efficacy, Generalization, Specificity, Fluency and Consistency, respectively.

| Method | Model | Counterfact | | | | | ZsRE | | |
|---|---|---|---|---|---|---|---|---|---|
| | | Eff.↑ | Gen.↑ | Spe.↑ | Flu.↑ | Consis.↑ | Eff.↑ | Gen.↑ | Spe.↑ |
| Pre-edited | | $7.85_{\pm0.26}$ | $10.58_{\pm0.26}$ | $89.48_{\pm0.18}$ | $635.23_{\pm0.11}$ | $24.14_{\pm0.08}$ | $36.99_{\pm0.30}$ | $36.34_{\pm0.30}$ | $31.89_{\pm0.22}$ |
| FT-L | Llama3 | $83.33_{\pm0.37}$ | $\underline{67.79}_{\pm0.40}$ | $46.63_{\pm0.37}$ | $233.72_{\pm0.22}$ | $8.77_{\pm0.05}$ | $30.48_{\pm0.26}$ | $30.22_{\pm0.32}$ | $15.49_{\pm0.17}$ |
| FT-W | | $61.23_{\pm0.38}$ | $62.40_{\pm0.24}$ | $47.05_{\pm0.41}$ | $492.34_{\pm0.23}$ | $3.57_{\pm0.03}$ | $32.08_{\pm0.35}$ | $\underline{31.43}_{\pm0.23}$ | $14.72_{\pm0.16}$ |
| MEND | | $63.24_{\pm0.31}$ | $61.17_{\pm0.36}$ | $45.37_{\pm0.38}$ | $372.16_{\pm0.80}$ | $4.21_{\pm0.05}$ | $0.91_{\pm0.05}$ | $1.09_{\pm0.05}$ | $0.53_{\pm0.02}$ |
| ROME | | $64.40_{\pm0.47}$ | $61.42_{\pm0.42}$ | $49.44_{\pm0.38}$ | $449.06_{\pm0.26}$ | $3.31_{\pm0.02}$ | $2.01_{\pm0.07}$ | $1.80_{\pm0.07}$ | $0.69_{\pm0.03}$ |
| MEMIT | | $65.65_{\pm0.47}$ | $64.65_{\pm0.42}$ | $51.56_{\pm0.38}$ | $437.43_{\pm1.67}$ | $6.58_{\pm0.11}$ | $34.62_{\pm0.36}$ | $31.28_{\pm0.34}$ | $18.49_{\pm0.19}$ |
| GRACE | | $\underline{90.72}_{\pm0.13}$ | $0.09_{\pm0.01}$ | $\underline{87.23}_{\pm0.21}$ | $\underline{632.43}_{\pm0.63}$ | $\underline{23.79}_{\pm0.23}$ | $\mathbf{74.58}_{\pm0.31}$ | $1.03_{\pm0.06}$ | $\underline{31.86}_{\pm0.12}$ |
| NSE | | $\mathbf{96.14}_{\pm0.19}$ | $\mathbf{78.42}_{\pm0.35}$ | $\mathbf{87.66}_{\pm0.19}$ | $\mathbf{632.72}_{\pm0.12}$ | $\mathbf{30.20}_{\pm0.10}$ | $\underline{62.29}_{\pm0.35}$ | $\mathbf{47.13}_{\pm0.31}$ | $\mathbf{32.32}_{\pm0.22}$ |
| Pre-edited | | $22.23_{\pm0.73}$ | $24.34_{\pm0.62}$ | $78.53_{\pm0.33}$ | $626.64_{\pm0.31}$ | $31.88_{\pm0.20}$ | $22.19_{\pm0.24}$ | $31.30_{\pm0.27}$ | $24.15_{\pm0.32}$ |
| FT-L | GPT2-XL | $63.55_{\pm0.48}$ | $42.20_{\pm0.41}$ | $57.06_{\pm0.30}$ | $519.35_{\pm0.27}$ | $10.56_{\pm0.05}$ | $37.11_{\pm0.39}$ | $33.30_{\pm0.37}$ | $10.36_{\pm0.17}$ |
| FT-W | | $42.70_{\pm0.49}$ | $35.93_{\pm0.40}$ | $63.06_{\pm0.31}$ | $565.96_{\pm0.23}$ | $13.03_{\pm0.06}$ | $24.97_{\pm0.32}$ | $22.40_{\pm0.30}$ | $12.73_{\pm0.18}$ |
| MEND | | $50.80_{\pm0.50}$ | $50.80_{\pm0.48}$ | $49.20_{\pm0.51}$ | $407.21_{\pm0.08}$ | $1.01_{\pm0.00}$ | $0.00_{\pm0.00}$ | $0.00_{\pm0.00}$ | $0.00_{\pm0.00}$ |
| ROME | | $54.60_{\pm0.48}$ | $51.18_{\pm0.40}$ | $52.68_{\pm0.33}$ | $366.13_{\pm1.40}$ | $0.72_{\pm0.02}$ | $47.50_{\pm0.43}$ | $43.56_{\pm0.42}$ | $14.27_{\pm0.19}$ |
| MEMIT | | $94.70_{\pm0.22}$ | $\underline{85.82}_{\pm0.28}$ | $60.50_{\pm0.32}$ | $477.26_{\pm0.54}$ | $22.72_{\pm0.15}$ | $79.17_{\pm0.32}$ | $71.44_{\pm0.36}$ | $\underline{26.12}_{\pm0.25}$ |
| GRACE | | $\underline{94.50}_{\pm0.24}$ | $0.04_{\pm0.01}$ | $\mathbf{78.13}_{\pm0.43}$ | $\underline{622.56}_{\pm0.79}$ | $\underline{31.55}_{\pm0.25}$ | $\underline{82.54}_{\pm0.21}$ | $0.40_{\pm0.02}$ | $24.78_{\pm0.21}$ |
| NSE | | $\mathbf{96.80}_{\pm0.20}$ | $\mathbf{87.72}_{\pm0.30}$ | $\underline{72.10}_{\pm0.28}$ | $\mathbf{622.85}_{\pm0.15}$ | $\mathbf{40.04}_{\pm0.11}$ | $\mathbf{83.26}_{\pm0.29}$ | $\mathbf{75.33}_{\pm0.34}$ | $\mathbf{26.14}_{\pm0.25}$ |
| Pre-edited | | $16.22_{\pm0.31}$ | $18.56_{\pm0.45}$ | $83.11_{\pm0.13}$ | $621.81_{\pm0.67}$ | $29.74_{\pm0.51}$ | $26.32_{\pm037}$ | $25.79_{\pm0.25}$ | $27.42_{\pm0.53}$ |
| FT-L | GPT-J | $92.15_{\pm0.27}$ | $72.38_{\pm0.38}$ | $43.35_{\pm0.37}$ | $297.92_{\pm0.77}$ | $6.65_{\pm0.10}$ | $72.37_{\pm0.29}$ | $68.91_{\pm0.32}$ | $19.66_{\pm0.23}$ |
| FT-W | | $48.35_{\pm0.49}$ | $31.42_{\pm0.39}$ | $68.71_{\pm0.28}$ | $587.20_{\pm0.23}$ | $29.41_{\pm0.09}$ | $39.81_{\pm0.36}$ | $32.55_{\pm0.33}$ | $27.76_{\pm0.26}$ |
| MEND | | $46.15_{\pm0.50}$ | $46.22_{\pm0.51}$ | $53.90_{\pm0.48}$ | $242.41_{\pm0.41}$ | $3.94_{\pm0.03}$ | $0.71_{\pm0.04}$ | $0.71_{\pm0.04}$ | $0.52_{\pm0.03}$ |
| ROME | | $57.50_{\pm0.48}$ | $54.20_{\pm0.40}$ | $52.05_{\pm0.31}$ | $589.28_{\pm0.08}$ | $3.22_{\pm0.02}$ | $56.42_{\pm0.42}$ | $54.65_{\pm0.42}$ | $9.86_{\pm0.16}$ |
| MEMIT | | $98.55_{\pm0.11}$ | $\mathbf{95.50}_{\pm0.16}$ | $63.64_{\pm0.31}$ | $546.28_{\pm0.88}$ | $\underline{34.89}_{\pm0.15}$ | $94.91_{\pm0.16}$ | $\underline{90.22}_{\pm0.23}$ | $27.56_{\pm0.27}$ |
| GRACE | | $\underline{95.88}_{\pm0.28}$ | $0.05_{\pm0.01}$ | $\mathbf{82.11}_{\pm0.24}$ | $\underline{620.21}_{\pm0.49}$ | $28.53_{\pm0.15}$ | $94.33_{\pm0.37}$ | $1.59_{\pm0.03}$ | $\underline{27.63}_{\pm0.43}$ |
| NSE | | $\mathbf{99.55}_{\pm0.06}$ | $\underline{91.92}_{\pm0.22}$ | $78.96_{\pm0.25}$ | $\mathbf{620.49}_{\pm0.16}$ | $\mathbf{40.24}_{\pm0.12}$ | $\mathbf{96.87}_{\pm0.14}$ | $\mathbf{91.33}_{\pm0.22}$ | $\mathbf{28.66}_{\pm0.25}$ |

## 4 EXPERIMENTS

We conduct experiments to demonstrate the effectiveness of our model editing method. The experiments aim to address the following research questions:

- **RQ1:** How does NSE perform on sequential model editing tasks compared to existing methods?
- **RQ2:** What is the impact of adjusting the batch size of edits on the performance of NSE?
- **RQ3:** Can the LLM, after undergoing NSE editing, retain its original general capabilities, and how does it perform on general capability tests?
- **RQ4:** How does each individual component of NSE contribute to the overall editing performance?

### 4.1 EXPERIMENTAL SETTINGS

**Datasets & Evaluation Metrics.** To evaluate the effectiveness of our method, we utilize two datasets: Counterfact (Meng et al., 2022) and ZsRE (Levy et al., 2017). For the Counterfact dataset, we employ five evaluation metrics as defined in previous work (Meng et al., 2022; 2023): **Efficacy** (efficiency success), **Generalization** (paraphrase success), **Specificity** (neighborhood success), **Fluency** (generation entropy), and **Consistency** (reference score). For the ZsRE dataset, we use three evaluation metrics also defined in previous work (Mitchell et al., 2022a; Meng et al., 2022; 2023): **Efficacy**, **Generalization**, and **Specificity**. For more details, see Appendix C.

**Models & Baselines.** Our comparative analysis evaluates the performance of various editing methods on three autoregressive language models, GPT2-XL (1.5B) (Radford et al., 2019), GPT-J (6B)

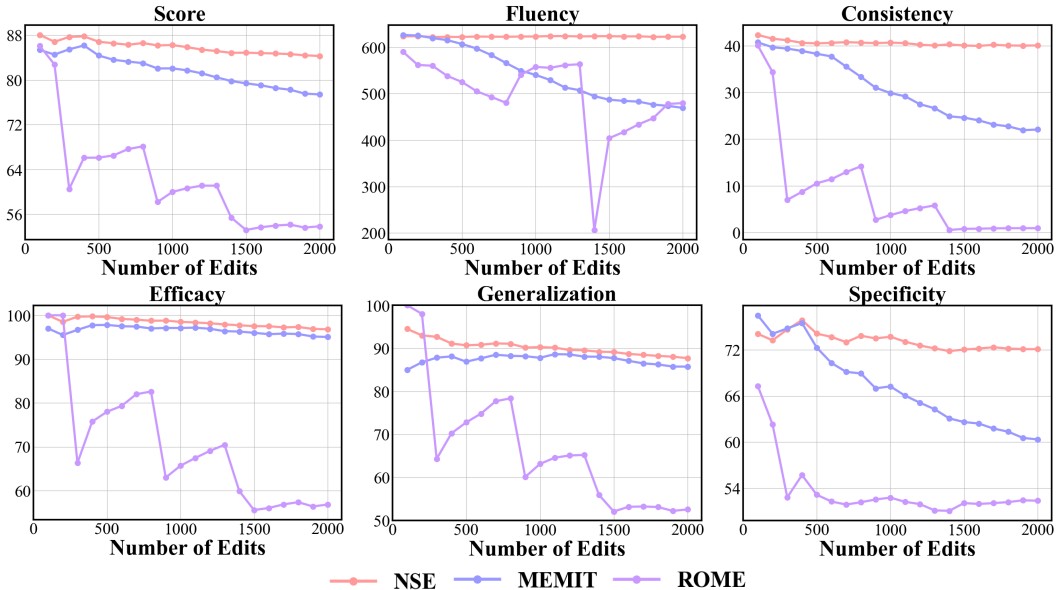

Figure 3: Editing performance of NSE and baselines with varying numbers of edits (batch size 100) in sequential editing, evaluated on the Counterfact dataset. Score is the harmonic mean of Efficacy, Generalization, and Specificity.

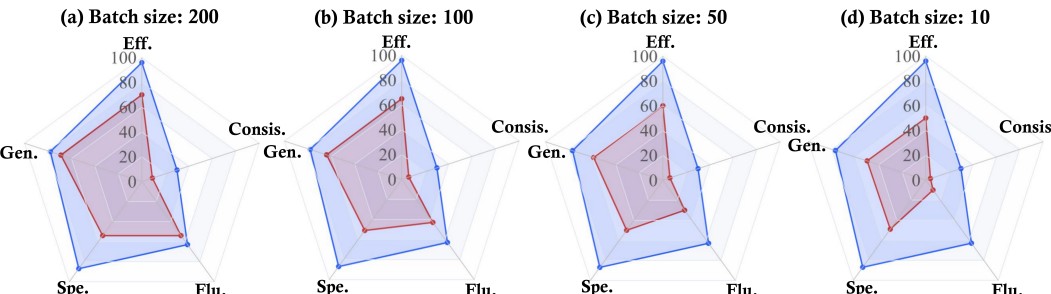

Figure 4: Editing performance of NSE and MEMIT with different batch size, evaluated on Llama3 (8B). The red line and the blue line represent MEMIT and NSE, respectively.

(Wang & Komatsuzaki, 2021) and Llama3 (8B)[1]. For baseline comparisons, we primarily select model editing methods that modify the model's parameters, including fine-tuning the specific layer (FT-L, FT-W) (Zhu et al., 2020), MEND (Mitchell et al., 2022a), ROME (Meng et al., 2022) and MEMIT (Meng et al., 2023). Additionally, we incorporated a memory-based editing method, GRACE (Hartvigsen et al., 2023), as a baseline. Although GRACE does not update model parameters and differs from our focus on parameter-modification editing methods, we chose to include it as a baseline due to its outstanding performance in sequential editing. Further details are provided in Appendix B.

## 4.2 PERFORMANCE COMPARISON (RQ1)

In this subsection, we provide a comprehensive comparison of NSE with existing methods on the sequential model editing task, using GPT2-XL, GPT-J and Llama3 models. The experiments are conducted with a total of 2000 edited samples and an editing batch size of 100 (batch size refers to the number of samples edited simultaneously in each editing round during the sequential editing process), evaluated on the Counterfact and ZsRE datasets. The results, under all evaluation methods, on all datasets, are presented in Table 1. Furthermore, we test the methods on the edited samples after each edit round on the GPT2-XL model using the Counterfact dataset. We presented the result on ROME, MEMIT and NSE in Figure 3. We also provide a case study comparing the text generation

---

[1]https://llama.meta.com/lama3/

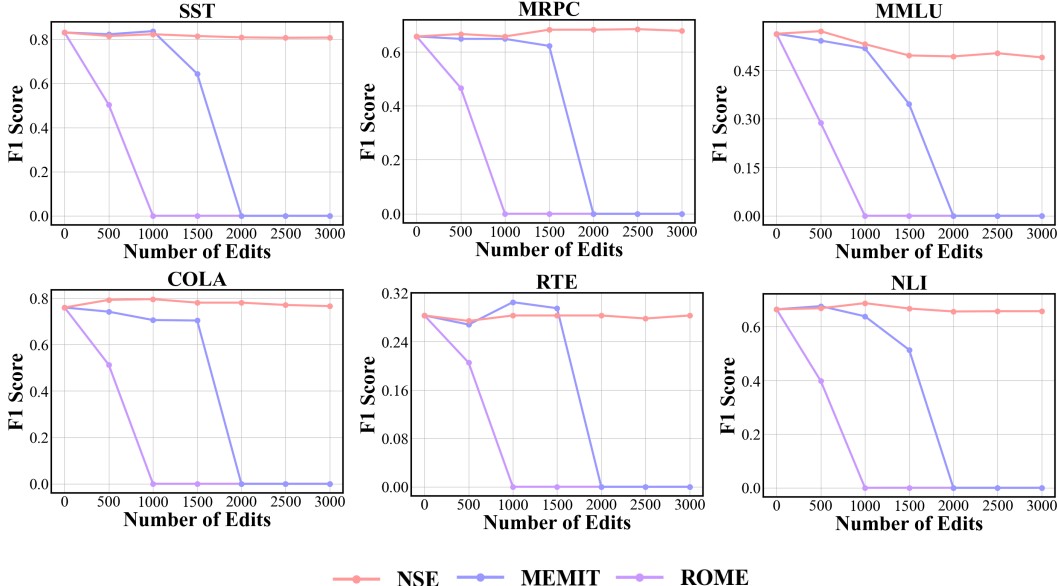

Figure 5: Performance on general tasks of edited models using NSE, ROME and MEMIT, with sequential editting on Llama3 (8B).

effects of different editing methods, with results available in Appendix F. According to these, we can find that:

- **Observation 1: NSE outperforms other baseline methods in almost all critical metrics across both datasets and models in the sequential editing task.** Specifically, compared to the parameter-modification baseline methods, NSE shows significant improvements across all metrics. Notably, on Llama3 (8B) editing, NSE achieves an average enhancement of around 30.33% across multiple metrics. Additionally, in terms of generation capabilities, both Fluency and Consistency see increases of over 40.75%. In contrast, while the parameter-preservation method GRACE retains model capabilities to the greatest extent, it exhibits weaker performance on the Generalization.

- **Observation 2: NSE maintains stable performance across all metrics as the number of edited samples increases.** As shown in Figure 3, NSE demonstrates robust performance, remaining resilient to model failure and forgetting despite the increase in the number of editing rounds. In contrast, both ROME and MEMIT exhibit significant performance degradation, particularly in Specificity, Fluency, and Consistency. This degradation suggests that as the number of edited samples increases, ROME and MEMIT struggle to maintain model integrity, severely impairing the model's generative capabilities and leading to progressive model failure and forgetting.

## 4.3 IMPACT OF BATCH SIZE (RQ2)

To answer RQ2, we examine the impact of different batch sizes on the performance of NSE compared to MEMIT, in the sequential model editing task, with a total of 2000 edits on the Counterfact dataset. Figure 4 presents four radar charts depicting the performance on Llama3 (8B) when using batch sizes of 200, 100, 50, and 10, respectively. We also tested the editing effects of GPT2-XL and GPT-J at different batch sizes, as detailed in Appendix G. According to Figure 4, we can find that:

- **Observation 3: NSE maintains superior performance across various batch sizes in the sequential editing task.** Specifically, as the batch size decreases, resulting in an increased number of editing rounds, the performance of MEMIT deteriorates. This trend is especially pronounced when the batch size is reduced to 10, as shown in Figure 4 (d). The radar charts reveal a significant decline in model editing effectiveness, with the notable decreases observed in all metrics. In contrast, NSE demonstrates an average improvement of 45.60% across these metrics.

Table 2: Ablation study results for NSE evaluated on GPT2-XL, GPT-J and Llama3 (8B).

| Method | Model | Eff.↑ | Gen.↑ | Spe.↑ | Flu.↑ | Consis.↑ |
|---|---|---|---|---|---|---|
| NSE | Llama3 | 96.14 | 78.42 | 87.66 | 632.72 | 30.20 |
| w/o weights rewinding | | 98.90 ↑2.76 | 91.18 ↑12.76 | 76.60 ↓11.06 | 625.65 ↓7.07 | 32.30 ↑2.10 |
| w/o neuron update | | 96.00 ↓0.14 | 77.13 ↓1.29 | 87.68 ↑0.02 | 632.68 ↓0.04 | 30.80 ↑0.60 |
| w/o iterative editing | | 95.65 ↓0.49 | 76.89 ↓1.33 | 87.58 ↓0.08 | 632.64 ↓0.08 | 30.24 ↑0.04 |
| NSE | GPT2-XL | 96.80 | 87.72 | 72.10 | 622.85 | 40.04 |
| w/o weights rewinding | | 96.90 ↑0.10 | 90.05 ↑2.33 | 61.73 ↓10.37 | 531.12 ↓91.73 | 31.07 ↓8.97 |
| w/o neuron update | | 96.30 ↓0.50 | 85.11 ↓2.61 | 73.08 ↑0.92 | 622.93 ↑0.08 | 40.84 ↑0.80 |
| w/o iterative editing | | 95.75 ↓1.05 | 86.31 ↓1.41 | 71.89 ↓0.21 | 622.63 ↓0.22 | 40.45 ↑0.41 |
| NSE | GPT-J | 99.55 | 91.92 | 78.96 | 620.49 | 40.24 |
| w/o weights rewinding | | 99.65 ↑0.10 | 94.98 ↑3.06 | 74.03 ↓4.93 | 615.22 ↓5.27 | 42.03 ↑1.79 |
| w/o neuron update | | 99.50 ↓0.05 | 91.52 ↓0.40 | 79.08 ↑0.12 | 620.63 ↑0.14 | 40.82 ↑0.58 |
| w/o iterative editing | | 98.65 ↓0.90 | 90.62 ↓1.30 | 77.90 ↓1.06 | 620.41 ↓0.08 | 40.45 ↑0.21 |

## 4.4 GENERAL ABILITY TEST (RQ3)

To assess the effects of model editing on the general capabilities of large language models (LLMs), we have selected six natural language tasks from the General Language Understanding Evaluation (GLUE) benchmark (Wang et al., 2019). The specific downstream tasks are as follows: (1) **SST (Stanford Sentiment Treebank)** (Socher et al., 2013), which involves classifying individual sentences extracted from movie reviews. (2) **MRPC (Microsoft Research Paraphrase Corpus)** (Dolan & Brockett, 2005), a benchmark for text matching and evaluating semantic similarity. (3) **MMLU (Massive Multi-task Language Understanding)** (Hendrycks et al., 2021), which assesses the multi-task accuracy of language models. (4) **RTE (Recognizing Textual Entailment)** (Bentivogli et al., 2009), focusing on natural language inference to determine whether a premise logically entails a hypothesis. (5) **CoLA (Corpus of Linguistic Acceptability)** (Warstadt et al., 2019), a single-sentence classification task using sentences derived from linguistic theory literature. (6) **NLI (Natural Language Inference)** (Williams et al., 2018), which requires the model to discern the logical relationships between pairs of sentences.

We conduct evaluations on Llama3 (8B) based on sequential editing settings with 3000 edits (batch size 100). The results are shown in the Figure 5. Here we can make the observations:

- **Observation 4: NSE consistently maintains the general capabilities of the LLM during sequential editing without incurring model failure.** Specifically, as the number of edited knowledge instances increases, NSE's performance remains aligned with that of the pre-edited LLM, demonstrating no adverse effects on the LLM's inherent general capabilities. In contrast, ROME and MEMIT exhibit a significant decline to 0 in general capabilities after editing approximately 1,000 to 2,000 samples, indicating that the model has already experienced degradation.

## 4.5 ABLATION STUDY (RQ4)

To assess the contributions of individual components in our method, we conduct an ablation study on the GPT-XL, GPT-J and Llama3 (8B) model using the Counterfact dataset. The results are presented in Table 2. We can make the observations as following:

- **Observation 5: Weights rewinding for value computation in NSE can effectively mitigate model failure.** Specifically, after the ablation of the weights rewinding component, there is a significant decline in both Specificity and Fluency for NSE, with an average decrease of 7.86%. Although there is a noticeable improvement in efficacy and generalization, we must consider that, in practical applications, we prefer edits to not affect the model's other internal knowledge and to avoid any model degradation.
- **Observation 6: Neuron-level weights updates and iterative multi-layer editing in NSE can effectively alleviate model forgetting.** Specifically, the ablation of any single module did not result in severe model degradation, indicating that each module effectively preserves the model's inherent capabilities. Particularly in terms of Efficacy and Generalization, the ablation of neuron-

level weights updates and iterative multi-layer editing leads to an average decrease of approximately $1\% - 2\%$, demonstrating that these modules further mitigate model forgetting.

## 5 RELATED WORK

Recent years have witnessed a development in methods for model editing, which are developed to adjust the behavior of LLMs within specific domains, while preserving their performance across other inputs. Current approaches to model editing in large language models (LLMs) generally fall into two main categories (Yao et al., 2023; Zhang et al., 2024b):

**Preserve Models' Parameters.** Methods that preserve the original model's parameters generally store edit examples in memory and use them to guide the model's predictions. For instance, SERAC (Mitchell et al., 2022b) keeps the original model unchanged and uses a separate counterfactual model for edits. T-Patcher (Huang et al., 2023) introduces an additional neuron for each output error, whereas CaliNet (Dong et al., 2022) incorporates knowledge using a predetermined number of neurons. Other methods like MemPrompt (Madaan et al., 2022) and IKE (Zheng et al., 2023) leverage the model's in-context learning by prompting it with edited facts and retrieved demonstrations from memory. MELO (Yu et al., 2024) dynamically activates corresponding LoRA blocks indexed in an inner vector database to alter the behavior of models. GRACE (Hartvigsen et al., 2023) implements sequential editing by maintaining a dynamically updated codebook. Larimar (Das et al., 2024) enhances LLMs with distributed episodic memory. OneEdit (Zhang et al., 2024a) introduces a neural-symbolic system for collaborative knowledge editing integrating knowledge graphs and LLMs.

**Modify Models' Parameters.** In contrast, methods that modify LLMs' parameters require updating the model's internal parameters with each edit. FT-W fine-tunes the specific layer with regularization constraints (Zhu et al., 2020). KN (Dai et al., 2022) identifies the "knowledge neuron" as the crucial pair in FFN and updates these neurons, which represents the knowledge. KE (Cao et al., 2021) and MEND (Mitchell et al., 2022a) use hypernetwork to forecast the weight change of LLMs based on the meta-learning methods. ROME (Meng et al., 2022) and MEMIT (Meng et al., 2023) support large-scale direct edits by locating and editing knowledge in specific layers of GPT. COMEBA-HK (Li et al., 2024) uses newly proposed hook layers to identify the editing scope supporting sequential editing. To enhance the performance of parameter modification methods in sequential editing, RECT (Gu et al., 2024) retains parameters with minimal changes to ensure stability, while PRUNE (Ma et al., 2024) constrains the maximum singular value of parameter changes. In this paper, we mainly focus on parameters-modification editing methods.

## 6 LIMITATION AND DISCUSSION

Despite the outstanding performance of NSE in sequential editing, our investigation reveals some limitations. Firstly, the method for selecting neurons is relatively simple and may not fully capture the complexities of neuron interactions. Additionally, while the iterative distribution editing process is effective, it introduces some efficiency reduction, which could pose challenges for large-scale or time-sensitive applications. Moving forward, we aim to explore more effective neuron attribution methods and enhance the efficiency of our editing techniques.

## 7 CONCLUSION

In this work, we introduce NSE, a model editing method for sequential model editing tailored for addressing the significant challenges of model forgetting and model failure. Specifically, we propose weights rewinding for value computation by optimizing the hidden states of the target layer using the model's original weights, which effectively minimize cumulative changes and maintains model coherence. Additionally, we select influential neurons for different knowlege to update weights in FFN and iteratively edit multi-layer weights, which effectively mitigate model forget. Experimental results on the GPT2-XL, GPT-J and Llama3-8B models using the Counterfact and ZsRE datasets demonstrate that NSE significantly outperforms existing model editing baselines and the ablation study shows that each component in NSE is effective.

## 8 ETHICS STATEMENT

Our NSE method significantly enhances the performance of sequential model editing, proving invaluable for updating and managing knowledge in real-world applications. While the capability to directly modify stored knowledge brings potential risks, such as the introduction of false or harmful information, we urge researchers to employ strict validation and oversight to ensure ethical use of these techniques. However, the original intent of model editing is positive, aiming to contribute to the efficient updates of large models in the future. Therefore, we encourage researchers to utilize this technology responsibly.

## 9 REPRODUCIBILITY

To ensure the reproducibility of our findings, detailed implementation instructions for NSE can be found in Appendix D. Additionally, the source code is available to the public at the following URL: https://anonymous.4open.science/r/NSE-0A8D/. These measures are taken to facilitate the verification and replication of our results by other researchers in the field.

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

# A  ALGORITHM

We present the NSE algorithm in sequential model editing task. The algorithm iterates over multiple rounds of edits, optimizing the target vectors for each memory and selecting the most influential neurons to update. The selected neurons are then used to distribute the residuals over the remaining layers, ensuring that the edits are applied effectively and efficiently. Detailed steps are provided in Algorithm 1.

---

**Algorithm 1** The NSE Algorithm

---

**Input:** Sequential edits $\{\mathbb{D}_{edit_t}\}_{t=1}^T = \{\{(s_i, r_i, o_i)\}_t\}_{t=1}^T$, given original LLM $f_{\theta_0}$, layers to edit
$\qquad \mathcal{R} = \{l_0 - |\mathcal{R}| + 1, \ldots, l_0\}$, covariances $\hat{\boldsymbol{C}}^{l^{-1}}$
**Output:** Modified generator containing edits from $\{\mathbb{D}_{edit_t}\}_{t=1}^T$

1: **for** each round $t = 1$ to $T$ **do**
2: $\quad$ **for** $(s_i, r_i, o_i) \in \mathbb{D}_{edit_t}$ **do**
3: $\qquad$ // Compute target $\mathbf{z}_i$ vectors for every memory $i$
4: $\qquad$ $\boldsymbol{z}_i \leftarrow \boldsymbol{v}_i^{l_0} + \boldsymbol{\delta}_i$
5: $\qquad$ Optimize $\boldsymbol{z}_i = \boldsymbol{v}_i^{l_0} + \arg\min_{\boldsymbol{\delta}_i}(-\log \mathbb{P}_{f_{\boldsymbol{\theta}_0}(\boldsymbol{v}_i^{l_0} += \boldsymbol{\delta}_i)} [o_i \mid (s_i, r_i)])$ $\qquad\qquad$ ▷ Eqn. 9
6: $\quad$ **end for**
7: $\quad$ // Select neurons for the current edits
8: $\quad$ **for** $(s_i, r_i, o_i) \in \mathbb{D}_{edit_t}$ **do**
9: $\qquad$ Compute $\boldsymbol{Q}_i = |\boldsymbol{k}_i|$ for layer $l$
10: $\qquad$ Select neurons $\mathcal{I}_i^l$ by
11: $\qquad$ $\mathcal{I} = \arg\min_{\mathcal{I} \subseteq \{1,\ldots,N\}} |\mathcal{I}| \quad$ s.t. $\quad \sum_{j \in \mathcal{I}} Q_{ij} \geq p \times \sum_{j=1}^N Q_{ij}$ $\qquad\qquad$ ▷ Eqn. 10
12: $\quad$ **end for**
13: $\quad$ **for** $l \in \mathcal{R}$ **do** $\qquad\qquad\qquad\qquad\qquad$ ▷ Perform update: spread changes over layers
14: $\qquad$ **while** $\mathbf{z}_i - \boldsymbol{h}_i^L > \alpha$ and maximum iterations not reached **do**
15: $\qquad\quad$ // Re-run the module
16: $\qquad\quad$ $\boldsymbol{h}_i^l \leftarrow \boldsymbol{h}_i^l + \boldsymbol{a}_i^l + \boldsymbol{v}_i^l$
17: $\qquad\quad$ Run layer $l$ with updated weights
18: $\qquad\quad$ **for** $(s_i, r_i, o_i) \in \mathbb{D}_{edit_t}$ **do**
19: $\qquad\qquad$ $\mathbf{r}_i^l \leftarrow \frac{\mathbf{z}_i - \boldsymbol{h}_i^L}{l_0 - l + 1}$
20: $\qquad\quad$ **end for**
21: $\qquad\quad$ $\hat{\boldsymbol{W}}^l \leftarrow \boldsymbol{W}^l[\mathcal{I}]$
22: $\qquad\quad$ $\hat{\boldsymbol{\Delta}}^l \leftarrow \boldsymbol{\Delta}^l[\mathcal{I}]$
23: $\qquad\quad$ $\hat{\boldsymbol{K}}_2^l \leftarrow \boldsymbol{K}_2^l[\mathcal{I}]$
24: $\qquad\quad$ Distribute residual over remaining layers
25: $\qquad\quad$ $\hat{\boldsymbol{\Delta}}^l \leftarrow \hat{\boldsymbol{R}}^l \hat{\boldsymbol{K}}_2^{l^T} \hat{\boldsymbol{C}}^{l^{-1}}$ $\qquad\qquad\qquad\qquad\qquad\qquad\qquad$ ▷ Eqn. 12
26: $\qquad\quad$ $\hat{\boldsymbol{W}}^l \leftarrow \hat{\boldsymbol{W}}^l + \hat{\boldsymbol{\Delta}}^l$ $\qquad\qquad$ ▷ Neuron-level updating layer $l$ MLP weights in model
27: $\qquad\quad$ Increment iteration counter
28: $\qquad$ **end while**
29: $\quad$ **end for**
30: **end for**

---

## B  BASELINES

The details of baselines are as follow:

- **FT-L** (Zhu et al., 2020) focuses on adjusting a specific layer identified by ROME (Meng et al., 2022), rather than fine-tuning all layers. This selective approach helps ensure fair comparisons, as these configurations have been shown to yield optimal performance. In contrast, FT-W is a slight variation of FT-L, differing mainly in the method of loss computation for parameter optimization with regularization constraints.
- **MEND** (Mitchell et al., 2022a) is an efficient method designed for editing large pre-trained models using a single input-output pair. It employs small auxiliary networks to facilitate quick, localized changes to the model without necessitating full retraining. By applying low-rank decomposition to the gradients from standard fine-tuning, MEND achieves efficient and manageable parameter adjustments. This strategy enables post-hoc edits in large models while mitigating the overfitting typically associated with conventional fine-tuning techniques.
- **ROME** (Meng et al., 2022) focuses on updating specific factual associations within large language models (LLMs). It identifies critical neuron activations within middle-layer feed-forward modules that influence factual predictions, allowing for direct modifications to the feed-forward weights. ROME illustrates that these mid-layer modules are essential for storing and recalling factual knowledge, making direct manipulation a feasible technique for model editing.
- **MEMIT** (Meng et al., 2023) is a scalable multi-layer update algorithm designed for efficiently incorporating new factual memories into transformer-based language models. Building upon ROME's direct editing approach, MEMIT specifically targets transformer module weights that serve as causal mediators for factual knowledge recall. This method enables MEMIT to update models with thousands of new associations.
- **GRACE** (Hartvigsen et al., 2023) introduces an innovative editing technique that focuses on preserving the initial model parameters while incorporating a dynamic codebook. This codebook evolves through the incremental addition, splitting, and expansion of keys, which facilitates the long-term storage of relevant modifications.

## C  DETAILS OF DATASETS AND EVALUATION METRICS

### C.1  DATASETS

ZsRE (Levy et al., 2017) is a question answering (QA) dataset that uses questions generated through back-translation as equivalent neighbors. Following previous work, natural questions are used as out-of-scope data to evaluate locality. Each sample in ZsRE includes a subject string and answers as the editing targets to assess editing success, along with the rephrased question for generalization evaluation and the locality question for evaluating specificity.

Counterfact (Meng et al., 2022) is a more challenging dataset that contrasts counterfactual with factual statements, initially scoring lower for Counterfact. It constructs out-of-scope data by replacing the subject entity with approximate entities sharing the same predicate. The Counterfact dataset has similar metrics to ZsRE for evaluating efficacy, generalization, and specificity. Additionally, Counterfact includes multiple generation prompts with the same meaning as the original prompt to test the quality of generated text, specifically focusing on fluency and consistency.

### C.2  ZSRE METRICS

Following the previous work (Mitchell et al., 2022a; Meng et al., 2022; 2023), this section defines each ZsRE metric given a LLM $f_\theta$, a knowledge fact prompt $(s_i, r_i)$, an edited target output $o_i$, and the model's original output $o_i^c$:

- **Efficacy**: Efficacy is calculated as the average top-1 accuracy on the edit samples:

$$\mathbb{E}_i \left\{ o_i = \arg\max_o \mathbb{P}_{f_\theta}(o \mid (s_i, r_i)) \right\}. \tag{13}$$

- **Generalization**: Generalization measures the model's performance on equivalent prompt of $(s_i, r_i)$, such as rephrased statements $N((s_i, r_i))$. This is evaluated by the average top-1 accuracy on these

$N((s_i, r_i))$:

$$\mathbb{E}_i \left\{ o_i = \arg\max_o \mathbb{P}_{f_\theta}(o \mid N((s_i, r_i))) \right\}. \tag{14}$$

- **Specificity**: Specificity ensures that the editing does not affect samples unrelated to the edit cases $O(s_i, r_i)$. This is evaluated by the top-1 accuracy of predictions that remain unchanged:

$$\mathbb{E}_i \left\{ o_i^c = \arg\max_o \mathbb{P}_{f_\theta}(o \mid O((s_i, r_i))) \right\}. \tag{15}$$

## C.3 Counterfact Metrics

Following previous work (Meng et al., 2022; 2023), this section defines each Counterfact metric given a LLM $f_\theta$, a knowledge fact prompt $(s_i, r_i)$, an edited target output $o_i$, and the model's original output $o_i^c$:

- **Efficacy (efficacy success)**: The proportion of cases where $o_i$ is more probable than $o_c^i$ with the $(s_i, r_i)$ prompt:

$$\mathbb{E}_i \left[ \mathbb{P}_{f_\theta}[o_i \mid (s_i, r_i)] > \mathbb{P}_{f_\theta}[o_c^i \mid (s_i, r_i)] \right]. \tag{16}$$

- **Generalization (paraphrase success)**: The proportion of cases where $o_i$ is more probable than $o_c^i$ in rephrased statements $N((s_i, r_i))$:

$$\mathbb{E}_i \left[ \mathbb{P}_{f_\theta}[o_i \mid N((s_i, r_i))] > \mathbb{P}_{f_\theta}[o_c^i \mid N((s_i, r_i))] \right]. \tag{17}$$

- **Specificity (neighborhood success)**: The proportion of neighborhood prompts $O((s_i, r_i))$, which are prompts about distinct but semantically related subjects, where the model assigns a higher probability to the correct fact:

$$\mathbb{E}_i \left[ \mathbb{P}_{f_\theta}[o_i \mid O((s_i, r_i))] > \mathbb{P}_{f_\theta}[o_c^i \mid O((s_i, r_i))] \right]. \tag{18}$$

- **Fluency (generation entropy)**: Measure for excessive repetition in model outputs. It uses the entropy of n-gram distributions:

$$-\frac{2}{3} \sum_k g_2(k) \log_2 g_2(k) + \frac{4}{3} \sum_k g_3(k) \log_2 g_3(k), \tag{19}$$

where $g_n(\cdot)$ is the n-gram frequency distribution.

- **Consistency (reference score)**: The consistency of the model's outputs is evaluated by giving the model $f_\theta$ a subject $s$ and computing the cosine similarity between the TF-IDF vectors of the model-generated text and a reference Wikipedia text about $o$.

# D    IMPLEMENTATION DETAILS

Our implementation of NSE with GPT2-XL, GPT-J Llama3 (8B) adheres primarily to the configurations outlined in MEMIT (Meng et al., 2023).

## D.1    IMPLEMENTATION DETAILS ON GPT2-XL

For GPT2-XL model, We target critical layers $[13, 14, 15, 16, 17]$ for editing. The matrix $\lambda \mathbb{E} \left[ \boldsymbol{kk}^T \right]$ is computed using 100,000 samples from Wikitext in fp32, with the hyperparameter $\lambda$ set to 20,000. During the process of computing $\mathbf{z}_i$, we perform 20 steps with a learning rate of 0.5. Additionally, we set the threshold $p$ for selecting neurons at 0.8. In the iterative distribution editing, we define a lower bound threshold $\alpha$ for $\|\mathbf{z}_i - \boldsymbol{h}_i^L\|^2$ as 35. Furthermore, we establish an upper bound of 150; if $\|\mathbf{z}_i - \boldsymbol{h}_i^L\|^2$ exceeds this upper limit, the sample is not edited to prevent the adverse effects of disabling edits on the model (Gupta et al., 2024a).

## D.2    IMPLEMENTATION DETAILS ON GPT-J

For GPT-J model, we target critical layers $[3, 4, 5, 6, 7, 8]$ for editing. The hyperparameter $\lambda$ is set to 15,000. During the process of computing $\mathbf{z}_i$, we perform 25 steps with a learning rate of 0.5. Additionally, we set the threshold $p$ for selecting neurons at 0.8. In the iterative distribution editing, we define a lower bound threshold $\alpha$ for $\|\mathbf{z}_i - \boldsymbol{h}_i^L\|^2$ as 15 and an upper bound as 100.

## D.3    IMPLEMENTATION DETAILS ON LLAMA3 (8B)

For Llama3 (8B) model, we target critical layers $[4, 5, 6, 7, 8]$ for editing. The hyperparameter $\lambda$ is set to 15,000. During the process of computing $\mathbf{z}_i$, we perform 25 steps with a learning rate of 0.1. Additionally, we set the threshold $p$ for selecting neurons at 0.8. In the iterative distribution editing, we define a lower bound threshold $\alpha$ for $\|\mathbf{z}_i - \boldsymbol{h}_i^L\|^2$ as 2.5 and an upper bound as 50.

## D.4    OTHER IMPLEMENTATION DETAILS

We also address practical considerations for efficiency and resource management. Specifically, when computing $\mathbf{z}_i$, we rely on the original model weights. To save space, we precompute $\mathbf{z}_i$ for the samples that will be edited in subsequent experiments and store these values. This approach allows us to call $\mathbf{z}_i$ directly during the editing process without needing to retain the original model weights, thereby optimizing storage requirements and computational efficiency.

All experiments are conducted on one A40 (48GB) GPU. The LLMs are loaded using HuggingFace Transformers (Wolf et al., 2019). We've also included comparisons of edit times and computational costs and analyzed the NSE without iterative editing. The results are presented in the Table 3.

Table 3: Times per edit for various methods evaluated on different models.

| Method | GPT2-XL | GPT-J | Llama3-8B |
|---|---|---|---|
| FT | 1.42s | 3.26s | 4.23s |
| FT-constrain | 1.44s | 3.74s | 4.35s |
| MEND | 0.12s | 0.13s | 0.13s |
| ROME | 2.57s | 4.82s | 5.73s |
| MEMIT | 2.51s | 4.74s | 5.54s |
| NSE | 3.21s | 5.51s | 6.23s |
| NSE-w/o iterative editing | 2.40s | 4.63s | 5.46s |

From the Table 3, it can be observed that the NSE method is slower than ROME/MEMIT. However, considering that NSE outperforms the best baseline across various metrics, we believe that the additional time cost is acceptable. Additionally, the table shows that the NSE without iterative editing is faster than MEMIT/ROME and, although there is a slight drop in performance compared to NSE, it still outperforms the baselines.

# E   ANALYSIS OF MULTI-LAYER EDITING APPROACH IN MEMIT

Firstly, we decompose the hidden state $h_i^{l_0}$ of the $l_0$-th layer in the Transformer architecture as follows:

$$h_i^{l_0} = h_i^l + \sum_{j=l}^{l_0} \left[ a_i^j(h_i^l) + v_i^j(h_i^l) \right], \qquad (20)$$

where $a_i^j(h_i^l)$ and $v_i^j(h_i^l)$ respectively denote the outputs of the attention and FFN layers at the $j$-th layer, given the input hidden state $h_i^l$ at layer $l$. Given that $\delta_i = z_i - h_i^{l_0}$, after applying the editing multi-layer algorithm at layer $l$ and assuming that the optimization in Eqn. 11 fits perfectly, the hidden state at layer $L$ is updated as:

$$h_i^{l_0} \leftarrow h_i^l + \frac{\delta_i}{l_0 - l + 1} + \sum_{j=l}^{l_0} \left[ a_i^j(h_i^l + \frac{\delta_i}{l_0 - l + 1}) + v_i^j(h_i^l + \frac{\delta_i}{l_0 - l + 1}) \right]. \qquad (21)$$

Substituting into Eqn. 20, and subtracting $z_i$ from both sides, we obtain:

$$\delta_i \leftarrow \frac{(l_0 - l)\delta_i}{l_0 - l + 1} + \sum_{j=l}^{l_0} \left[ a_i^j(h_i^l) - a_i^j(h_i^l + \frac{\delta_i}{l_0 - l + 1}) + v_i^j(h_i^l) - v_i^j(h_i^l + \frac{\delta_i}{l_0 - l + 1}) \right]. \qquad (22)$$

If we ignore the effects of the attention and FFN layers and the errors in the fitting process, the residual $\delta_i$ before updating layer $l$ can be recursively calculated as $\frac{(l_0 - l + 1)\delta_i^{(0)}}{|\mathcal{R}|}$, where $\delta_i^{(0)}$ represents the initial residual before any layers are edited in one round. Consequently, the update $v_i$ at $l$-th layer ($l \in \mathcal{R}$) can be expressed as $v_i^l += \frac{\delta_i^{(0)}}{|\mathcal{R}|}$, conceptually distributing the total change $\delta_i^{(0)}$ uniformly across all layers targeted for editing. Each layer's edit still nudges $h_i^{l_0}$ closer to $z_i$, but the ignored errors mean a single execution of the editing distribution algorithm often fails to sufficiently approximate $h_i^{l_0}$ to $z_i$.

# F CASE STUDY

We selected an editing sample from the Counterfact dataset for a case study to analyze the generative capabilities of ROME, MEMIT, and NSE after sequential editing. This case study was conducted on the GPT2-XL, GPT-J and Llama3 model after performing sequential editing with 2000 total editing samples and a batch size of 100. The results are shown in Table 4, Table 5 and Table 6. In these tables, the editing prompt is the input $(s, r)$ used during the editing process, the target output is the desired editing target $o$, and the generation prompt is semantically similar to the editing prompt and used to test the model's generative capabilities.

The results demonstrate that ROME failed to include the target output "Romania" in its generation, and the model's output became incoherent and unreadable. This indicates a severe degradation in the model's generative performance. MEMIT, although successful in editing, produced output that repeatedly mentioned the target "Romania", which also signifies a model failure. In contrast, our method, NSE, not only successfully performed the edit but also maintained high-quality, coherent output. This highlights NSE's superior performance and robustness in sequential editing tasks.

Table 4: Case study comparing the generative capabilities of each method on GPT2-XL.

| Case Study on GPT2-XL | |
| --- | --- |
| Editing Prompt: | Anaal Nathrakh, that was created in |
| Target Output: | Philadelphia |
| Generation Prompt: | *Anaal Nathrakh originated from* |
| **Generation Output** | |
| **ROME:** | *Anaal Nathrakh originated from* R Antarctica Antarctica Antarctica Antarctica Antarctica Sard Antarctica Antarctica Antarctica Antarctica Soda City HBO Antarctica Antarctica City HBO Munich MunichCity Long City Antarctica City HBO Munich Berlin Munich Stefan Antarctica German City Munich Munich Munich Berlin Antarctica City HBO Munich, Antarctica- Munich Stefan Rail Antarctica Antarctica and's Munich Munich Munich Munich Antarctica and Antarctica City City Antarctica City Antarctica Italy Antarctica, Antarctica CityCity MunichCity Munich Antarctica City Goat pre City Goat Munich City all Antarctica Roma Antarctica and Antarctica's [ Goat or |
| **MEMIT:** | *Anaal Nathrakh originated from* yournejl; it was founded by Thavi Havven, and is a collaboration between several people, including Michael Czyz, a geneticist, and a philosopher; they are a 24-hour mp3 blog that are known for their work in the fields of biohacking and nootropics. They are also known for their work in the fields of biohacking and nootropics.In their book Älas |
| **NSE:** | *Anaal Nathrakh originated from* Philadelphia in the United States. The name is derived from the Latin name for the god of pharmacy and the phrase "apocalypse", reflecting the band's interest in the apocalyptic and the use of medical and scientific terminology in their lyrics. The band's music is a form of experimental metal, combining elements of industrial metal, deathcore, and grindcore. Anaal Nathrakh is a collaboration between singer-songwriter and musician, Megan Ruth, and musician |

Table 5: Case study comparing the generative capabilities of each method on GPT-J.

| Case Study on GPT-J | |
|---|---|
| Editing Prompt: | Lower Assam is located in the country of |
| Target Output: | Romania |
| Generation Prompt: | *Lower Assam's surroundings include* |
| **Generation Output** | |
| ROME: | *Lower Assam's surroundings include* newly Nordic, Antarctica, Munich (Antarctica, Antarctica, London, safe Munich,,'s closest London,,ring's mechanisms Antarctica, Area Sea, Manchester seemingly,'s wholes'redund, flatass nestingeri,inc presum agenda, presumety Indian bellig's mechanisms closest,secret transientCam sign wholes elements redund nearest, closest's nature other's safeeth scar near, closest near,ET's pred's close emet |
| MEMIT: | *Lower Assam's surroundings include* Romania, Romania, Romania, Romania The Romanian language, Romania, Romania, Romania, Romania, Romania, Romania, Romania, Romania, Romania, Romania, Romania, Romania, Romania, Romania, Romania, Romania, Romania, Romania, Romania, Romania, Romania, Romania, Romania, Romania, Romania, Romania, Romania, Romania. Romanian, Romanian, Romanian, Romania, Romania, Romania, Romania. Romanian, Romanian, Romanian, Romania. Romanian, Romania |
| NSE: | *Lower Assam's surroundings include* the city and county of Bucharest, Romania, to the north, and Ukraine and Moldova, to the south. To the west is the Black Sea. Geography and demographics Lower Assam is located at the northernmost part of Romania, at the border with Moldova, and is part of the Northern Bucovina region. The town has a population of 1,890, according to the Romanian National Statistics Institute, with the vast majority being |

Table 6: Case study comparing the generative capabilities of each method on Llama3 (8B).

| Case Study on Llama3 | |
|---|---|
| Editing Prompt: | The mother tongue of Danielle Darrieux is |
| Target Output: | English |
| Generation Prompt: | *Danielle Darrieux's mother tongue is* |
| **Generation Output** | |
| **ROME:** | *Danielle Darrieux's mother tongue is* ,- Municharius City Antarctica Antarctica Antarctica Antarctica Antarctica Goat City HBO Antarctica Antarctica Antarctica's's City HBO- Port City Antarctica, Antarctica-, Antarctica City Antarctica Antarctica Antarctica Goat HBO Antarctica Munich Munich Stefan Antarctica in Antarctica Italy Antarctica Antarctica Antarctica City ( Munich MunichCity Lay Antarctica Antarctica Antarctica- Munich Munich Munich Stefan Goat fishes,, Antarctica Antarctica Antarctica Antarctica- same Antarctica Antarctica Sard and, Munich Munich Munich Stefan Il Munich Munich Antarctica's City ( R cont |
| **MEMIT:** | *Danielle Darrieux's mother tongue is* Poleled Russian ( French French Paris French French Manchester Berlin Parisaggiled French English English English French French French French French French Italian German Paris French French French French French Italian French French French English Paris German French French French French Russian Paris French French French French French English German Russian Alexander Russian Russian Russian French French French Russian French Italian Italian Italian French French Russian Italian Russian French French French Russian Jerome Paris French Italian Russian French Italian Russian French Russian Russian Italian French French Italian, |
| **NSE:** | *Danielle Darrieux's mother tongue is* English. Danielle Darrieux was born in London, England, UK. She has English and French ancestry. Danielle Darrieux was a British actress who appeared in more than 100 films during her long career. Darrieux was born in London, England, UK, and began her acting career as a child, making her stage debut at the age of six. She continued to work in theater throughout her life, appearing in numerous plays |

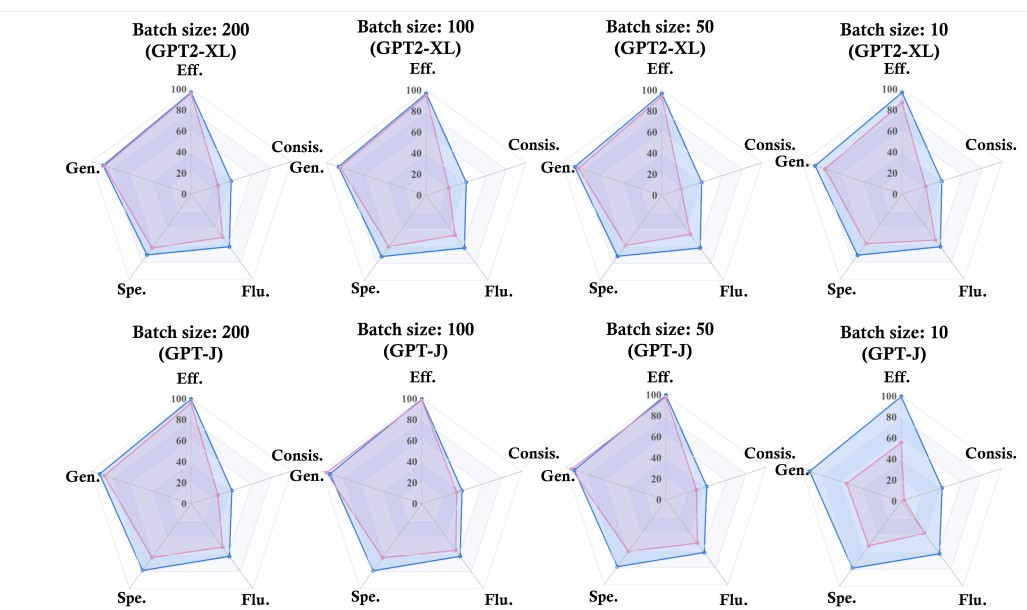

Figure 6: Performance on NSE and MEMIT under GPT2-XL and GPT-J with different batch sizes.The red line and the blue line represent MEMIT and NSE, respectively

Table 7: 2000 sequential editing samples with different neuron selection thredhold $p$ on GPT-J

| Thredhold $p$ | Counterfact | | | | | ZsRE | | |
|---|---|---|---|---|---|---|---|---|
| | Eff.↑ | Gen.↑ | Spe.↑ | Flu.↑ | Consis.↑ | Eff.↑ | Gen.↑ | Spe.↑ |
| 0.85 | $99.50_{\pm0.07}$ | $91.28_{\pm0.21}$ | $77.82_{\pm0.25}$ | $619.25_{\pm0.17}$ | $40.82_{\pm0.12}$ | $96.80_{\pm0.14}$ | $\mathbf{92.19}_{\pm0.21}$ | $28.14_{\pm0.25}$ |
| 0.8 | $\mathbf{99.55}_{\pm0.06}$ | $\mathbf{91.92}_{\pm0.22}$ | $78.96_{\pm0.25}$ | $\mathbf{620.49}_{\pm0.16}$ | $40.24_{\pm0.12}$ | $\mathbf{96.87}_{\pm0.14}$ | $91.33_{\pm0.22}$ | $\mathbf{28.66}_{\pm0.25}$ |
| 0.75 | $99.45_{\pm0.07}$ | $91.68_{\pm0.22}$ | $\mathbf{79.03}_{\pm0.24}$ | $620.42_{\pm0.16}$ | $\mathbf{40.83}_{\pm0.12}$ | $96.80_{\pm0.14}$ | $91.66_{\pm0.22}$ | $27.68_{\pm0.25}$ |

# G  MORE QUANTITATIVE RESULTS

We provide more detailed experimental results. Figure 6 presents the results of our method, NSE, compared to the baseline MEMIT on GPT2-XL and GPT-J, under different batch sizes in sequential editing, with a total of 2000 editing samples.

Additionally, Table 7 shows the performance of NSE with different neuron selection thresholds $p$. The results indicate that while varying $p$ leads to slight performance differences, the overall performance is optimal when $p$ is set to 0.8.

## H    VISUALIZING THE ZSRE AND COUNTERFACT DATASETS THROUGH EXAMPLES

To facilitate a better understanding of model editing tasks for readers who may be new to this field, we present two examples from the Counterfact and ZSRE datasets in Figure 7 and 8. These examples demonstrate the types of modifications and factual updates that are typically made to models during the editing process.

```
{
    "case_id": 0,
    "pararel_idx": 2796,
    "requested_rewrite": {
        "prompt": "The mother tongue of {} is",
        "relation_id": "P103",
        "target_new": {
            "str": "English",
            "id": "Q1860"
        },
        "target_true": {
            "str": "French",
            "id": "Q150"
        },
        "subject": "Danielle Darrieux"
    },
    "paraphrase_prompts": [
        "Shayna does this and Yossel goes still and dies. Danielle Darrieux, a native",
        "An album was recorded for Capitol Nashville but never released. Danielle Darrieux spoke the language"
    ],
    "neighborhood_prompts": [
        "The mother tongue of L\u00e9on Blum is",
        "The native language of Montesquieu is",
        "Fran\u00e7ois Bayrou, a native",
        "The native language of Raymond Barre is",
        "Michel Rocard is a native speaker of",
        "Jacques Chaban-Delmas is a native speaker of",
        "The native language of Fran\u00e7ois Bayrou is",
        "Maurice Genevoix, speaker of",
        "The mother tongue of Fran\u00e7ois Bayrou is",
        "Melchior de Vog\u00fc\u00e9, speaker of"
    ],
    "attribute_prompts": [
        "J.\u00a0R.\u00a0R. Tolkien is a native speaker of",
        "The mother tongue of Douglas Adams is",
        "The mother tongue of Paul McCartney is",
        "Elvis Presley is a native speaker of",
        "Barack Obama, speaker of",
        "Douglas Adams, speaker of",
        "Meryl Streep, a native",
        "George Orwell spoke the language",
        "George Washington, a native",
        "Michael Jackson, a native"
    ],
    "generation_prompts": [
        "Danielle Darrieux's mother tongue is",
        "Where Danielle Darrieux is from, people speak the language of",
        "Danielle Darrieux's mother tongue is",
        "Danielle Darrieux was born in",
        "Danielle Darrieux's mother tongue is",
        "Danielle Darrieux's mother tongue is",
        "Danielle Darrieux was born in",
        "Where Danielle Darrieux is from, people speak the language of",
        "Danielle Darrieux was born in",
        "Danielle Darrieux was born in"
    ]
}
```

Figure 7: A Sample of the Counterfact dataset.

```
{
    "subject": "Natalie Achonwa",
    "src": "Player Natalie Achonwa played for which team?",
    "pred": "Washington Mystics",
    "rephrase": "Which team is Natalie Achonwa in?",
    "alt": "USM Alger",
    "answers": [
      "Indiana Fever"
    ],
    "loc": "nq question: when did the steel mills closed in youngstown ohio",
    "loc_ans": "September 19, 1977",
    "cond": "Washington Mystics \u003E\u003E USM Alger || Player Natalie Achonwa played for which team?"
},
{
    "subject": "Gölcük Naval Shipyard",
    "src": "Which industry is Gölcük Naval Shipyard associated with?",
    "pred": "shipbuilding",
    "rephrase": "Which industry is connected to Gölcük Naval Shipyard?",
    "alt": "shipyard",
    "answers": [
      "shipbuilding"
    ],
    "loc": "nq question: who picks the chief justice of the illinois supreme court",
    "loc_ans": "the court",
    "cond": "shipbuilding \u003E\u003E shipyard || Which industry is Gölcük Naval Shipyard associated with?"
},
{
    "subject": "Konrad Barde",
    "src": "What is the date of death for Konrad Barde?",
    "pred": "16 July 1882",
    "rephrase": "What's the death date for Konrad Barde?",
    "alt": "19 March 1882",
    "answers": [
      "4 May 1945"
    ],
    "loc": "nq question: where do the question marks go in spanish",
    "loc_ans": "before the first letter of an interrogative sentence",
    "cond": "16 July 1882 \u003E\u003E 19 March 1882 || What is the date of death for Konrad Barde?"
},
{
    "subject": "Pierre Corneille",
    "src": "What is the language of Pierre Corneille?",
    "pred": "French",
    "rephrase": "Which language does Pierre Corneille have?",
    "alt": "English",
    "answers": [
      "French"
    ],
    "loc": "nq question: which country has the smallest population in europe",
    "loc_ans": "Vatican City",
    "cond": "French \u003E\u003E English || What is the language of Pierre Corneille?"
}
```

Figure 8: A Samples of the ZsRE dataset.

