# OpenReview forum: "Neuron-Level Sequential Editing for Large Language Models"
_ICLR.cc/2025/Conference — ICLR 2025 Conference Withdrawn Submission_

### Official Review · Reviewer_ZcGG · 2024-10-26

**Soundness:** 2
**Presentation:** 2
**Contribution:** 2
**Rating:** 3
**Confidence:** 3

**Summary:**

This work proposes a framework including weight rewinding, neuron-level updating, and iterative editing for improving the efficacy of sequence knowledge editing while preserving the model utility. The experiments on editing performance, batch size of edits, and preservation of utility, as well as ablation study demonstrate the effectiveness of the proposed method.

**Strengths:**

- The sequential editing problem is essential for enabling the model's lifelong learning capability.
- The authors conduct comprehensive experiments to demonstrate the superiority of their method.

**Weaknesses:**

- There are some claims that are not supported by any references or experiments. For example, in line 199-201, the authors claim that "This indicates that the cumulative parameter updates from each editing round can lead to a shift in value computation. Conversely, using the original model parameters fθ0 to compute zi effectively prevents this issue" without any supports. And in line 255-257, where the claim is "However, due to errors in the fitting process, some knowledge proves difficult to edit, resulting in vl0 i not sufficiently approximating zi, and consequently leading to editing failures." The authors do not define "errors" and they do not provide references for their claim.
- The neuron-level weights updating belongs to localization-informed methods, which is widely used in knowledge-editing-related tasks [1,2]. Here I give examples of privacy information editing, which I believe are similar to knowledge editing.
- The iterative multi-layer editing module will cause inefficiency in the knowledge editing process.
- The experiment setting is inconsistent. In section 4.2, the number of edited samples is 2000, while in section 4.4, the number is 3000. The authors do not explain the reason why they use different numbers of edited samples in the paper.

[1] Fan C, Liu J, Zhang Y, et al. SalUn: Empowering Machine Unlearning via Gradient-based Weight Saliency in Both Image Classification and Generation[C]//The Twelfth International Conference on Learning Representations.

[2] Wu X, Li J, Xu M, et al. DEPN: Detecting and Editing Privacy Neurons in Pretrained Language Models[C]//Proceedings of the 2023 Conference on Empirical Methods in Natural Language Processing. 2023: 2875-2886.

**Questions:**

- I do not understand the equation 9. Can you explain how you preserve the weights and the operation in previous works in detail?
- In the experiment, the authors set the number of edited samples as 2000. However, in MEMIT, the number is 10000. I am wondering if the proposed method still performs well when processing more samples.
- In the neuron-level weights updating, does the selected neuron always change? Is there any specific neuron being edited many times?
- In the section 4.3, why do you only choose MEMIT for comparison? I find GRACE is also a strong baseline and it is more persuasive if you add GRACE in the section 4.3.
- In the ablation study, I suggest experiments with two modules removed to find out which module plays the most important role in the proposed method.

---

### Official Review · Reviewer_pBoX · 2024-11-03

**Soundness:** 3
**Presentation:** 2
**Contribution:** 3
**Rating:** 5
**Confidence:** 4

**Summary:**

This paper proposes a method called Neuron-level Sequential Editing (NSE) based on MEMIT to extend that method to the sequential model editing task. The main component is to always compute latent updates based on the original model weights, in order to avoid model failures during the sequential editing. They also conduct neuron-level update and iterative editing to further improve the performance. The comprehensive experiments show the superiority of their proposed method compared to editing methods that directly modify the model’s parameters.

**Strengths:**

1. This paper focuses on the sequential editing task and the editing methods that modify the original model’s parameters. They extend the previous baseline MEMIT to this scenario and prove the superiority of their proposed method.
2. The authors have done comprehensive experiments to show the superiority of their method on the sequential model editing task compared to methods that modify the model’s parameters.
3. The authors release their code for reproducibility.

**Weaknesses:**

1. The authors should make it clearer in their presentation that they focus on methods that modify the parameters. The models that freeze the model’s parameters, e.g., SERAC[1] and T-Patcher[2], show stable performance in sequential editing[3]. The authors don’t compare their method with these two strong baselines in the experiments. Therefore, it is more appropriate to clearly state this constraint in the Introduction, especially in the first paragraph. The first paragraph constrains the model editing methods into “Modify model’s parameters” methods (ROME and MEMIT), but neglect other methods that can also address this problem. In the following paragraphs, authors use *direct model editing* methods (which is not clear what this stands for) and *memory-based* methods to refer to *modify models’ parameters* and *preserve model’s parameters* methods (the taxonomy used in the Related Work section) respectively. It would be clearer and more coherent if using the taxonomy in the Related Work section across the entire paper.
2. The presentation of the mathematical notations should be improved. Details follow in the Questions section.
3. The main architecture figure 2 is misleading. $z_i$ is not a weight of the model, but a hidden state. The decoder blocks are not frozen since the W_out is updated.
4. It is not clear what “number of edits” means in the figures. To me, one edit means one edit to the model, i.e., one update to the model. However, it seems that “number of edits” refers to “number of edit samples” in this paper. In the previous papers, number of model edits and number of edit samples are proportional, since they only use once the edit samples. It’s not the case in the NSE as the authors adopt an iterative editing technique where one edit sample can be used to edit several times the model.

[1] Mitchell, Eric, et al. "Memory-based model editing at scale." International Conference on Machine Learning. PMLR, 2022.

[2] Huang, Zeyu, et al. "Transformer-Patcher: One Mistake Worth One Neuron." The Eleventh International Conference on Learning Representations. 2023

[3] Yao, Yunzhi, et al. "Editing Large Language Models: Problems, Methods, and Opportunities." Proceedings of the 2023 Conference on Empirical Methods in Natural Language Processing. 2023.

**Questions:**

1. Issues related to mathematical notations.
    1. In section 2.1, the vectors and matrices should be italic to be consistent with other sections.
    2. In line 145, $n$ is not defined.
    3. In line 150, $\Delta, W$ and $V_1$ are not bolded.
    4. In line 219, it is not clear that the activation values are from a single layer or from all the layers to be modified. If it’s the activation values of a certain layer, it would be better to add the subscript of the layer $l$.
    5. In line 220, Q should be italic. In addition, the absolute value of a vector is its norm. I think here the authors want to say the absolute value for each element in the vector. It would be clearer that the authors add some clarifications about what this absolute value sign means.
    6. In equation 10, it is not appropriate to use the same symbol on both sides of the equation, one refers to a specific set while the other refers to a variable. It is thus not clear which set is referred to in the inequality.
    7. In equations 3, 4, and 5, “edit” should be non-italic.
    8. In Figure 2, the “in” and “out” for MLP layer weights should be non-italic, which will be consistent with the notation in line 230.
2. Does the batch size influence the edit success? I’m curious whether using different batch sizes will result in different number of model edits.
3. How probable is it that the method cannot edit successfully one sample and needs to do it iteratively?
4. In Figure 5, the degradation seems very sudden. Do the authors have any explanation why this degradation is sharp rather than gradual.
5. In line 431, what is the baseline model for the average improvement of 45.6%?
6. Although it seems out of the scope of this paper (depending on how the authors will address the weaknesses point 1), it would be interesting to compare the NSE method with T-Patcher and SERAC.

---

### Official Review · Reviewer_qqEa · 2024-11-04

**Soundness:** 3
**Presentation:** 3
**Contribution:** 2
**Rating:** 5
**Confidence:** 3

**Summary:**

This paper introduces Neuron-level Sequential Editing (NSE), a method for sequentially editing LLMs' internal knowledge. Traditional single-edit methods often struggle with model forgetting and failures in multi-edit scenarios. NSE tackles these issues by optimizing neuron-level hidden states while preventing failures. NSE reduces forgetting and outperforms existing editing methods.

**Strengths:**

1. Quantitative experiments are adequate and effective.
2. The methodology is clear and well-understood.

**Weaknesses:**

More discussion and ablation analysis are required, please see the Questions.

**Questions:**

1. Could you explain the reasons for the varying performance of NSE, ROME, and MEMIT in Fig. 5?
2. Please discuss the trade-offs between Specificity, Fluency, and Efficacy for the GPT-XL, GPT-J, and Llama3 models using the Counterfact dataset.
3. Could you propose specific methods to optimize the computation time based on the result in Tab. 3?

---

### Note · Authors · 2024-12-16

I have read and agree with the venue's withdrawal policy on behalf of myself and my co-authors.